# *Low phytic acid* Crops: Observations Based On Four Decades of Research

**DOI:** 10.3390/plants9020140

**Published:** 2020-01-22

**Authors:** Victor Raboy

**Affiliations:** USDA-ARS Small Grains and Potato Research Unit, 1691 South 2700 West, Aberdeen, ID 83210, USA; vraboy@gmail.com

**Keywords:** phytic acid, phosphorus, phytate, low phytic acid, lpa, seed, genetics, animal nutrition, human nutrition, sustainability

## Abstract

The *low phytic acid* (*lpa*), or “low-phytate” seed trait can provide numerous potential benefits to the nutritional quality of foods and feeds and to the sustainability of agricultural production. Major benefits include enhanced phosphorus (P) management contributing to enhanced sustainability in non-ruminant (poultry, swine, and fish) production; reduced environmental impact due to reduced waste P in non-ruminant production; enhanced “global” bioavailability of minerals (iron, zinc, calcium, magnesium) for both humans and non-ruminant animals; enhancement of animal health, productivity and the quality of animal products; development of “low seed total P” crops which also can enhance management of P in agricultural production and contribute to its sustainability. Evaluations of this trait by industry and by advocates of biofortification via breeding for enhanced mineral density have been too short term and too narrowly focused. Arguments against breeding for the low-phytate trait overstate the negatives such as potentially reduced yields and field performance or possible reductions in phytic acid’s health benefits. Progress in breeding or genetically-engineering high-yielding stress-tolerant low-phytate crops continues. Perhaps due to the potential benefits of the low-phytate trait, the challenge of developing high-yielding, stress-tolerant low-phytate crops has become something of a holy grail for crop genetic engineering. While there are widely available and efficacious alternative approaches to deal with the problems posed by seed-derived dietary phytic acid, such as use of the enzyme phytase as a feed additive, or biofortification breeding, if there were an interest in developing low-phytate crops with good field performance and good seed quality, it could be accomplished given adequate time and support. Even with a moderate reduction in yield, in light of the numerous benefits of low-phytate types as human foods or animal feeds, should one not grow a nutritionally-enhanced crop variant that perhaps has 5% to 10% less yield than a standard variant but one that is substantially more nutritious? Such crops would be a benefit to human nutrition especially in populations at risk for iron and zinc deficiency, and a benefit to the sustainability of agricultural production.

## 1. Introduction: The *low phytic acid* Trait

Phytic acid (myo-inositol-1,2,3,4,5,6-hexakisphospate; Figure 1A) is the storage form of phosphorus (P) in seeds, typically representing from 75 ± 10% of seed total P [1]. Following synthesis during seed development, it accumulates and is deposited as mixed “phytate” or “phytin” salts primarily of potassium (K) and magnesium (Mg). These salts may also contain iron (Fe) and zinc (Zn) and other mineral cations. Prior to 1980 (the year I began my PhD studies of genetic and environmental factors impacting soybean phytic acid), there was no “Mendelian” genetics of seed phytic acid; there were no known mutations or variant alleles that perturbed seed phytic acid synthesis or accumulation. The first *low phytic acid* (*lpa*) mutants of any crop species, maize (*Zea mays* L.) *low phytic acid* 1-1 (*Zmlpa*1-1) and *Zmlpa*2-1, were isolated around 1990 in my then fairly new USDA-ARS (Agricultural Research Servic) lab [2]. Homozygosity for these first two maize mutations reduced seed phytic acid (myo-inositol hexakisphosphate) by 50% to 70% while having little discernible effect on seed total phosphorus (P) (illustrated in Figure 1B for maize *lpa*1-1). Isolation of maize *lpa*1-1 and *lpa*2-1 was followed in my lab and many others with the isolation of mutations in a third maize locus [3], *lpa* mutants in barley (*Hordeum vulgare* L.), rice (*Oryza sativa* L.), wheat (*Triticum aestivum* L.), soybean [*Glycine max* (L.) Merr.], common bean (*Phaseolus vulgaris* L.), in several additional crops and model systems such as *Arabidopsis thaliana* (L. (Heynh)), and increasingly via genetic engineering in a number of crops [4,5,6]. In seed homozygous for most but not all *lpa* mutations or genotypes, reductions in phytic acid P are largely matched by increases in inorganic P, with little or no change in seed total P. In some cases (such as maize *lpa*2) reductions in phytic acid P result in increased inorganic P and increases in other inositol phosphates such as myoinositol tris-, tetra*kis*, or penta*kis*phosphate, but the seed total P remains similar to wild-type (data not shown).

Here, I will present what I think are a few interesting observations and views about the practical or “applied” side of seed phytic acid and the potential role of *lpa* crops, first in reference to phytic acid’s role in animal and human nutrition and second in reference to questions about breeding or engineering the low-phytate trait. The two main problems that the *lpa* trait can be used to address are (1) P management in non-ruminant animal production and; (2) mineral deficiency in humans. For many years there have been attractive alternative approaches and technologies to address these problems. For the P management issue, there is the use of phytase (phytic acid-specific phosphohydrolases; myo-inositol hexakisphosphate 3- and 6-phosphohydrolase; EC 3.1.3.8 and EC 3.1.3.26) as a feed additive [7,8] and for the human mineral nutrition problem, there is “biofortification” breeding for enhanced mineral density [9]. Both of these alternatives are less challenging to implement and more readily accessible than breeding the *lpa* trait and perhaps in specific cases have advantages over it, resulting in much wider acceptance, utilization and recognition. Does the *lpa* approach have a useful place in addressing these two problems? Does the *lpa* trait possibly have any advantages over these alternative approaches? 

I will address these questions and present views developed through the course of nearly four decades of work. In summary, I have two main observations. First, in regards to the “low-phytate” trait and the alternative approaches to the problems it presents in human health and animal agriculture, people tend to “miss the forest for the trees”. People tend to look at or address isolated effects or problems, but do not consider the bigger picture which is the sum of the positive effects described below. For example, studies might address the effect of reducing dietary phytate on iron retention but not on zinc, calcium or magnesium retention. Or biofortification breeding programs might target iron-dense crops, or zinc-dense crops, but usually not both in the same germplasm. Few if any have addressed what I define as “global” mineral nutritional quality; the nutritional quality of food in terms of all the major mineral nutrients. Programs or studies might address the problem of P management in animal production or the problem of mineral deficiency in humans, but not both. Second, I believe people’s efforts or views are often too short-term, such that one might miss the long-term goals and benefits. For example, the development of a high-yielding low-phytate crop might require long-term, sustained effort in breeding or genetic engineering. This sort of long-term effort is not typical of most institutions. 

First, an estimate of the potential value of the low-phytate trait will provide a basis for subsequent discussion. This will include a brief review of the role seed phytic acid plays as a major bottleneck in the flow of P through the world’s agricultural ecosystem. Next, the fundamentally important question of the role of phytic acid in animal and human nutrition will be addressed. Last, a few points regarding the breeding the low-phytate crops will be addressed.

## 2. An Estimate of the Potential Value of the *low phytic acid* Trait

Studies conducted over the years with *lpa* mutants and genotypes of maize and other species indicate that the positive impact of this single-gene or simply-inherited trait on seed nutritional quality compares very favorably with any other single- or multigenic trait described to-date. As a preliminary example, in the case of maize *lpa*1-1 anecdotal evidence (unpublished) from researchers in the early 1990s indicated that pigs whose diet consisted solely of *lpa*1-1 maize, with no additional supplementation, did fairly well. They did not display the severe P and mineral deficiency and poor bone density observed if they were solely fed wild-type maize, which has low bioavailable P. The single *lpa*1-1 mutation erased these deficiencies and bone disease, converting maize into a nearly complete food. Can one imagine a situation where it might be beneficial to have a staple food and feed grain crop that is a nearly complete food, requiring little additional supplementation? 

Seed phytic acid P was recognized as early as 1939 as a “non-available” form of P for monogastric animals (poultry, swine, fish) [10,11,12]. In the 20th century, there was growing concern over the contribution of phytic acid-derived P in animal waste to water pollution and eutrophication [13]. So, as the first major benefit of the *lpa* trait, studies utilizing *lpa* lines (reviewed in [6]) revealed that use of these lines in non-ruminant animal feeds enhanced seed-derived P bioavailability proportional to the decrease in seed phytic acid P and increase in inorganic P, and depending on diet formulation, could substantially reduce animal waste P. 

Since the mid-20th century, seed-derived dietary phytic acid was well known to play an important role in the nutritional quality of human foods, via its negative impact on mineral (iron, zinc, magnesium, calcium, phosphorus) retention and utilization [14,15]. Mineral deficiency, especially iron and zinc deficiency, was widely recognized as a major international public health problem [16,17]. As the second major benefit of the *lpa* trait, studies indicated that human and non-ruminant mineral nutrition including iron, zinc, and calcium nutrition, is enhanced following consumption of foods or feeds prepared with *lpa* types as compared with normal-phytate types (reviewed in [6]). 

Perturbation of phytic acid synthesis or accumulation may also favorably alter the distribution of minerals across the tissues of the cereal grain, in some cases resulting in higher mineral levels in the central starchy endosperm, in turn resulting in higher mineral density in milled products like white rice (*Oryza sativa* L.; reviewed in [6]). It is also likely that protein utilization will be enhanced by reductions in seed-derived dietary phytic acid [7]. The biosynthetic pathways to both phytic acid and the raffinosaccharide series of sugars share a common precursor, myo-inositol [18]. Knock-out of the soybean (*Glycine max* L. (Merr.)) genome’s seed-specific myo-Inositol 1-phosphate synthase (“MIPS”) gene, encoding the enzyme which is the sole biosynthetic source of the myo-inositol ring, resulted in seeds with both reduced phytic acid and reduced raffinosaccharides, the later an undigestible and undesirable component of foods [18]. 

A study published in 2000 found that pigs consuming *lpa*1-1 diets as compared with normal-phytate diets were leaner, had enhanced muscle density and had less “backfat” [19]. A patent was then awarded to a reputable company claiming a method to produce eggs with reduced overall cholesterol and enhanced cholesterol quality (reduced relative levels of low-density lipoproteins), this method being consumption of maize *lpa1-1* as compared with wild-type [20]. Since crop lines with varying, genetically-determined levels of endogenous seed phytic acid were not available prior to the isolation of these first *lpa* mutants, one might predict that their subsequent use in nutrition studies would yield unexpected findings or outcomes not anticipated based on prior science. The finding of the potential benefit of the low-phytate trait in cholesterol management represents an early example of this. Mechanisms that might contribute to enhanced “leanness” and muscle density, desirable traits in humans as well as pigs, include the possibility that consumption of *lpa* feeds leads to enhanced overall mineral nutritional health, enhanced P/Ca status, and enhanced protein digestion resulting in enhanced amino-acid nutritional status. 

Any discussion of real-world problems relating to seed phytic acid should include the fact that it represents a major bottleneck in P flux through the world-wide agricultural ecosystem (Figure 2). The total amount of seed phytic acid P annually produced by major crops represents a sum equivalent to nearly 65% of fertilizer P manufactured annually worldwide [21]. This bottleneck represents a potentially valuable target in efforts to reduce the negative environmental impact of agricultural production. Agricultural P runoff contributes to surface water pollution and eutrophication, which in turn leads to oxygen depletion, die-off and “dead zones” [22]. This is a major, ongoing, world-wide and newsworthy (https://www.nytimes.com/interactive/2019/12/25/world/europe/farms-environment.html) problem. For example, a recent survey [23] found that in terms of eutrophication, greater than 95% of the Baltic sea is considered a “problem area”. Low-phytate genetics can help with the global eutrophication problem in two ways: (1) first and foremost via its beneficial change in seed chemistry where total P remains unchanged but substantially more of that P is bioavailable for non-ruminants, resulting in more P in “product” and less P in “waste” (Figure 2); (2) via *lpa* alleles that both alter seed chemistry and condition reduced seed total P amount. For example, if one could reduce seed P by 20% but not impact yield, that would be equivalent to increasing the “fuel efficiency” of that crop, at least in terms of the macronutrient P; one would obtain the same amount of grain per unit of production but during harvest remove or ”mine” 20% less P from the field, leaving it in the field for subsequent years of production [24]. 

That is exactly the case with barley *lpa*1-1 [25,26,27] (reviewed in [24]). In addition to a ~50% reduction in grain phytic acid P, it also conditions a 15% to 20% reduction in grain total P, while having little or no effect on crop yield (see below). Thus barley *lpa*1-1 represents the first low-seed total P crop variant necessary for this novel approach to enhancing P management in crop production. In fact, it conditions a seed P phenotype that represents the ideal for all end uses: reduced seed total P accompanied by reduced phytic acid P. 

It has estimated that for annual global rice production alone, a 20% reduction in soil P mining during crop production, achieved via a genetic reduction of a similar extent in seed total P, could save producers “several hundred million US dollars annually in fertilizer inputs” [28]. It would also help address a potentially even more important long-term problem, “Peak Phosphorus” (reviewed in [22]). Phosphorus used for fertilizer is obtained from a potentially limiting resource, rock phosphate. Future reserves may for both political and technical reasons become limiting to agricultural production. A 20% reduction in phosphate use that has little impact on crop productivity could significantly enhance the long-term sustainability of agricultural production. 

Barley *lpa*1-1′s good field performance and yield has led to the breeding and subsequent release of two barley cultivars with this unique seed P phenotype; “Herald” [29] and “Clearwater” [30]. The release of such cultivars represents a very solid proof-of-principle and validation of the feasibility of both (1) the “low-phytate” approach to enhancing the nutritional value of crops and (2) the “low-seed total P” approach to enhancing P management in agriculture. 

The gene perturbed in barley *lpa*1-1 was subsequently identified as encoding a member of the sulfate family of transporters and thus termed *HvST (H. vulgare Sulfate Transporter*; [31]). Some members of this large gene family have functions other than sulfate transport. Thus the ortholog perturbed in barley *lpa*1-1 probably functions in P transport specific to phytic acid synthesis. This possibility was confirmed via analysis of a rice ortholog of this gene, termed SULT-like Phosphorus Distribution Transporter (SPDT; [32]). A knock-out of this rice ortholog resulted in a seed P phenotype almost identical to that of barley *lpa*1-1, reduced phytate and a 20% to 30% reduction in seed total P, and like barley *lpa*1-1, appeared to have little impact on plant performance in an initial field trial. 

The work with the barley *lpa*1-1 and the rice the SPDT knockouts also illustrate an important value of the classical “forward genetics” approach represented by the isolation of *lpa* mutants. Forward genetics is defined as the isolation of a mutant phenotype followed by identification of the mutation and gene conditioning that phenotype. That value is that it provides the information needed to then conduct “reverse genetics”, where perturbations of that gene are first obtained and the phenotypes they condition subsequently described. In the case of the low-phytate trait, the first example of this process was the isolation of maize *lpa1-1* [2], which then led to the identification of the gene perturbed in maize *lpa*1-1, the sequence of which was then subsequently used to genetic engineer the trait into soybean. [33]. 

As a summary of this estimate of the potential value of the *low phytic acid* trait: taken together, these various nutritional and P-management benefits, of value to human nutrition and animal production, and to efforts to reduce the environmental impact and enhance the sustainability of agricultural production, are pretty amazing for a single-gene or simply-inherited seed chemistry trait. 

In 1987, when starting my career in the USDA-ARS, it was helpful to have the positive rationale that isolating *lpa* mutants would be useful in breeding of *lpa* crops, which in turn could be used to reduce water pollution from animal agriculture, then a popular concern. However, I viewed breeding low-phytate crops as a long-term and even speculative goal. I had two other initial goals that I thought were more straightforward in the short-term: to use such genetics resources to study the biology of phytic acid in plants, such as figuring out phytic acid’s biosynthetic pathway; to use the near-isogenic lines we could develop to better analyze the role of seed phytic acid in human and animal nutrition. In the 1980′s it was not yet widely understood that inositol hexaphosphate is the most abundant inositol phosphate in nature, and a central metabolic pool in the eukaryotic cell [34,35]. The “inositol phosphate pathways” are central to cellular sensing and signaling including P sensing and signaling [4], and several other critically important cellular processes in all eukaryotes, such as RNA processing. In this context, the first maize *lpa* mutants were essentially the first inositol phosphate pathways mutants identified in any eukaryotic species. The isolation of these and other *lpa* mutants has led to the identification of several novel genes and functions and thus has contributed to our understanding of the biosynthesis of phytic aid during seed development and has contributed to basic cell biology [3,4,33,36,37]. For example, the maize *lpa*1 gene was the first inositol hexaphosphate transporter identified in any species, and such transporters are key to the roles inositol hexaphosphate plays in cellular metabolism and signaling [33]. 

But here I will mostly focus on the applied side of things, the role of seed phytic acid in animal and human nutrition and the breeding/genetic engineering of *lpa* crops.

## 3. Lessons Learned From Animal and Human Nutrition Studies 

Working for four decades in the field of genetics, breeding, and engineering food and feed crops for enhanced nutritional value gives one some perspective. Below I will attempt to explain why: (1) I think that in the case of the use of the *lpa* trait in animal feeds, decisions were made early on that focused too much on short-term profitability and too much solely on P management and; (2) in breeding or genetics to enhance nutritional value of crops for human foods, too great a focus was placed on biofortification via enhanced mineral density to the detriment of support for breeding aimed at antinutrients like phytic acid. In both the feed and food cases, these decisions then impacted long-term strategies and long-term funding support possibly to the detriment of agricultural production’s efficiency and sustainability, and environmental and public health. This is only an opinion and I, of course, could be biased or wrong, or perhaps simply just “venting” about what I perceive as the lack of support for the low-phytate approach. 

### 3.1. The Low Phytic Acid Trait and “Available Phosphorus” in Non-Ruminant Animal Agriculture

First let us take the example of how the *lpa* trait was initially viewed and evaluated by an Ag biotech company when it was first available for use in crop breeding in the early 1990s. The main interest in seed phytic acid in the agricultural industry in the U.S. and Europe was and largely remains its role as the major P fraction in grains and legumes destined for use in poultry and swine feeds. Since phytic acid P is “non-available P” whereas essentially all other forms of P are “available P” (Figure 3A; reviewed in [6]), the low-phytate trait in maize was initially referred to as “High Available P” or “HAP” corn [38]. The approach initially used by this Ag biotech company to determine the value of the *lpa* trait was in terms of the dollar value of the increase in available P, using then-current (early 1990s) P prices. For example, using the market cost for rock phosphate (https://www.indexmundi.com/commodities/?commodity=rock-phosphate&months=300); if an *lpa* allele converts half of the phytic acid P in seeds from “non-available P” to “available P”, in 1990 the available P in a low-phytate line would represent ~$5 billion (US) (Figure 3B, right), as compared with $1.66 billion (US) for the available P in the “normal phytate” line (Figure 3B, left). 

Since P was relatively inexpensive in 1990, valuating the low-phytate trait in terms of “feed P equivalents” created several hurdles to development and production. Rather than using the well-proven “penetration pricing” strategy, where new products are initially sold at relatively low prices to capture market share [39,40], industry participants at that time wanted to attach a “technology surcharge” to HAP hybrid seed, in my opinion, a short-sighted business strategy. The problem was the “feed P value” of the first *lpa* types, based on market prices for P in the early 1990s, was not sufficient to accommodate this “technology surcharge”. But what if the price of P for use in feeds and fertilizers dramatically increased over the next 10 to 20 years? What if the price of P will continue to increase or experience price spikes? It takes a minimum of ten but often many more years to go from the initial development of a new biotech trait to its widespread marketing [41]. Should not that early decision been based on future projections of P prices? Today the increasing cost and potential future scarcity of rock phosphate for use in fertilizer P and feed P production is changing views of the importance of seed phytic acid. By 2015, the cost of rock phosphate had roughly tripled since 1990, so that net available P in a low-phytate line where half the phytate is converted to available P would represent $15 billion (US) (Figure 3B, right). Also, using this same estimate of the dollar value of seed P, if the 20% to 30% reduction in seed total P trait of barley *lpa*1-1 or the rice SPDT knock-outs were engineered into all major grain and legume seed crops, that could potentially represent a savings of up to$ 4.5 billion (US), annually. 

A second way to look at the value of the seed P represented by phytic acid P (non-available P) is to consider the total market value for the feed additive phytase needed to break down a given amount of feed phytic acid (Figure 3B, middle). In this discussion, it is useful to remember that the world market for phytase as a feed additive is the largest market for any industrial enzyme. In 2015, the total market value for the standard rate of the feed additive phytase (500 units enzyme/kilo feed, an application rates designed to breakdown ~50% of feed-derived phytic acid), was >$500 million (US). The objective of this application rate is not to improve animal health or productivity, but rather to meet regulatory standards designed to ameliorate water pollution by dictating acceptable levels of animal waste P. Subsequently, participants in the industry started advocating phytase “superdosing”; application rates twice, three times or more than 500 units per kilo feed (http://phytate.info/superdosing-phytase). The primary rationale for phytase superdosing was that the resulting application rates achieved a near-complete breakdown of dietary phytic acid and that this had added benefits beyond the 50% improvement in dietary P utilization achieved via the “500 enzyme unit” application rate. These additional benefits were enhanced overall animal health, resulting largely from optimized mineral nutritional health, leading to enhanced productivity. A secondary benefit was more efficient use of increasingly expensive P and optimized reduction in waste P. If a proposed “superdosing” use of phytase becomes industry standard, this would then add at a minimum another >$500 mMillion (US) (Figure 3B, middle). 

So if annual phytase costs were used to value a genetically-determined 50% reduction in seed phytic acid, it would at a minimum be worth $0.5 Billion for maize alone. But who gets that value? Is it the grain grower, the phytase producer, or the livestock producer? If the dollar value of enhancing P management and reducing P waste is captured in the grain, breeding low-phytate crops should ultimately bring profits to the grain grower. 

The rationale for phytase superdosing was based in part on the findings of the first generation of animal nutrition studies using the initial maize and barley *lpa* near-isogenic lines my lab USDA-ARS lab-produced: genetically-determined reductions in crop seed phytic acid translated into enhanced mineral nutritional health in a global sense (enhanced P, Ca, Zn, Fe, Mg etc.), and in other possible benefits such as enhanced protein utilization, that in sum resulted in healthier, more productive animals. This validated one of my major initial objectives in studying seed phytic acid genetics: that the resulting development of sets of near-isogenic lines that largely differ only in seed phytic acid P level will be valuable in studying the role of phytic acid in animal and human nutrition. Thus the resulting research was not only of value in studying the role of dietary phytic acid but also led to rationalizing increased phytase use! 

That an opportunity was missed in the 1990s by the maize breeding industry is highlighted by the fact that little known subsequent studies have shown that both corn farmers who might grow HAP corn and consumers of foods produced using the non-GMO HAP corn favorably viewed its environmental benefits [42,43]. A survey of the Delmarva Peninsula (on the eastern side of the Chesapeake Bay, US) corn farmers found a willingness to grow non-GMO HAP corn due to the environmental benefits but increasing resistance if doing so was accompanied by higher production costs or reduced yield [42]. A separate study found consumers looked favorably on chicken produced using HAP corn [43], again due to the environmental benefits. This also serves as a good example of how focusing just on P management limits a full appreciation of the value of the *lpa* trait. What if those farmers or consumers were instead asked if they would look favorably on use of a new corn type in feeds that are “high available P”, with the resulting environmental benefits, but also results in healthier animals due to a global enhancement of mineral nutritional health and possibly healthier animal products, due to reduced cholesterol and increased “leanness”? 

One critical hurdle to the adoption of HAP corn was that initial yield studies of lines into which an *lpa* allele had been crossed indicated that it was associated with a yield loss of perhaps 5% to 10%. Since *lpa* corn is competing with commodity “yellow-dent” corn for the end-use strictly as a feed ingredient, in the horizontally-organized agricultural system in the U.S., where grain growers in most cases derive their income entirely from the amount of grain they can produce, that initial *lpa*-associated yield loss represented a financial hurdle. There was little consideration at the time of how sustained breeding and research, meaning over five to 10 years or longer, might result in elite performance of a non-GMO *lpa* type. That is simply too great a time span for any private-sector company’s balance sheet. Further, in these initial deliberations in the early 1990′s, consideration of environmental issues or sustainable management of P issues were secondary to the simple question of the cost of feed P. No predictions of possible changes in feed P costs over the long-term, meaning over decades, were included. 

These ways of estimating the value of the *lpa* trait are meant only to provide a historical perspective on views and attitudes that impact decision making in the agricultural industry. Looking at this trait simply in terms of the value of the P in seeds as determined by market rates for P fertilizer or feed P undervalues the trait over the longer term, and does not take into account sustainability, the value of reduced environmental impact and water quality, nor the potential full benefits for animal health and productivity independent of P nutrition. 

### 3.2. Low Phytic Acid Near-Isogenic Lines: A Powerful Model to Study the Impact of Seed-Derived Dietary Phytic Acid in Human and Animal Nutrition

The power of *lpa* near-isogenic lines as an experimental tool in human and animal research is that one can produce test foods or feeds with very accurately known levels of seed-derived endogenous phytic acid that are stable and vary from wild-type levels in a step-wise fashion, through moderate reductions to lines which produce seed with a near-absence of phytic acid. No additional experimental (“artificial”) manipulations are needed. As a result, when using these lines, the impact of dietary phytic acid on some aspect of nutrition can be assayed more quantitatively, and with a higher degree of accuracy, than is possible using other experimental approaches. Furthermore, any difference in nutritional outcomes observed between wild-type versus *lpa* near-isogenic lines can be attributed to the single-gene allelic difference conditioning the trait. As an illustration, consider two nutrition studies, one with pigs (*Sus scrofa domesticus* (Erxleben)) and one with trout (*Oncorhynchus mykiss* (Walbaum)), that were conducted with a set of four barley near-isogneic lines (Figure 4) [44,45]. These four lines produce seed with either wild-type levels of seed phytic acid (cv. Harrington), or seed with step-wise reductions in phytic acid: a ~40% reduction in *Hvlpa*1-1, a ~70% reduction in *Hvlpa*3-1, and a >95% reduction in *Hvlpa*M955. This provides a nicely linear set of feed phytic acid treatments that in both studies revealed a highly linear negative correlation between grain phytic acid level and calcium bioavailability (“% apparent digestibility coefficient” or (ADC) in the trout study and “% retention/intake” in the pig study).

Calcium nutritional health is both important to animal health and productivity and is a significant public health issue worldwide, impacting at least 200 million people, many of whom consume diets rich in phytic acid. A study with human subjects confirmed the findings in studies with pigs or trout (Figure 4): that genetically-determined reductions in seed phytic acid (in this case maize) translated into enhanced calcium bioavailability [46]. Tortillas were prepared using seed produced by a pair of maize near-isogenic “Dent Corn” hybrids: a “wild-type” hybrid that produced seed with wild-type levels of phytic acid, or *Zmlpa*1-1 hybrid that produced seed with a ~66% reduction in phytic acid. “Fractional calcium absorption” was 43% greater following consumption of the *Zmlpa*1-1 tortillas compared with that observed following consumption of “wild-type” tortillas.

One important conclusion arrived at via observation of linearity of response has to do with the historical use of thresholds of dietary phytic acid in studying its impact on iron and zinc bioavailability. While defining such thresholds is important if not essential to progress in understanding the impact of dietary phytic acid, a large number of studies utilizing wild-type and *lpa* isolines in various crop species, and conducted with both human subjects and animal models, clearly illustrates that genetically-determined, step-wise reductions in grain- or seed-derived dietary phytic acid typically results in what appears to be fairly linear increases in iron and zinc bioavailability. For example, an oft-quoted paper established a widely-accepted threshold that a reduction in dietary phytic acid of at least 90% was essential to see benefits for iron bioavailability [47]. But studies with maize *lpa* isolines revealed that iron absorption was 49% greater in human subjects following consumption of foods prepared with maize grain with reductions in phytic acid of only 66%, as compared with normal-phytate grain [48]. 

In the case of zinc, reductions in the phytic acid:zinc molar ratio to <10 to 20 (depending on levels of phytic acid, zinc, calcium, and other dietary constituents) have been viewed as necessary to observe positive impacts in zinc bioavailability [49,50]. An initial study with human subjects that evaluated a pair of “near-isogenic” maize hybrids, one a “normal-phyate” hybrid and the second an *lpa* hybrids, found that the reduction of the phytic acid:zinc molar ratio from 36 in the normal-phytate grain to 17 in the *lpa* grain nearly doubled fractional zinc absorption (from 0.17 to 0.30) [51]. A second study using two sets of near-isogenic hybrids reported a linear negative relationship between phytic acid:zinc molar ratios (ranging from 7.5 to 35), and fractional zinc absorption [52]. While the phytic acid:zinc molar ratios in both low-phytate types was <20, what is important is that there appeared to be a linearity of response [52]. In addition to the importance these observations have for public policy and the development of strategies to address mineral nutritional health, they also should inform the selection of targets for breeding *lpa* crops: even moderate reductions as compared with wild-type would result in enhanced iron and zinc bioavailability.

### 3.3. “Biofortification” via Breeding for Elevated Zinc or Iron vs. the Low Phytic Acid Approach? Has the Harvest Plus Project Been Too One-Sided? Has it Missed an Opportunity to Support a Type of Crop Breeding that would Benefit Human Health?

Biofortification via breeding crops for elevated micronutrient density is an effective approach to addressing micronutrient deficiency in at-risk populations, both in terms of crop genetics but also as an “appropriate technology”, the latter defined in part as a technology that is practical for and suitable to the social and economic conditions of specific locale, culture or society [53,54]. In recognition of the importance of this strategy to enhancing human health in at-risk populations through crop breeding, one of its main advocates, Howarth Bouis of the International Food Policy Research Institute (IFPRI), was recognized in 2016 with the World Food Prize (https://www.ifpri.org/news-release/howarth-bouis-wins-world-food-prize). As of 2017, it is estimated that 20 million people in farm households in the developing world were growing biofortified crops [54]. That is impressive. This was in part the result of the vision and the ultimately successful acquisition of support and funding that did not come without struggle and substantial persistence (https://www.harvestplus.org/about/our-history). 

I’ve often wondered though that this success may have resulted in “paradigm block” and pushed other approaches to the sidelines. It may have resulted in the “canalization” of funding. But in the case of mineral deficiency, if one considers not just iron and zinc but other important mineral nutrients such as calcium and magnesium, is breeding for elevated levels of micronutrients going to be efficacious if phytic acid levels remain high? In at least one case, the biofortification breeding of common bean (*Phaseolus vulgaris* L.) for elevated iron levels, the short answer is no [55]. I will return to these particular studies and the recommendations that resulted from them in more detail below.

If the goal is to enhance mineral nutritional health globally (to enhance mineral status for all nutritionally-important minerals), can one simultaneously breed or engineer high levels of many of the most nutritionally-important minerals? The argument has been made that in some cases selection or breeding for elevated levels of one element may also result in elevated levels of other elements because there are positive correlations between the seed concentrations of some elements [56]. This may occur in some cases, but I find arguments for this approach to understate its complexities. For example, seed calcium levels are not well correlated with zinc or iron levels, and the distribution of calcium, iron, and zinc in the grain differ greatly, impacting levels in food products made from differing grain fractions. In contrast, the results of animal and human trials of low-phytate types described here indicate that reducing dietary phytic acid via the low-phytate approach would result in enhanced global mineral nutrition.

Furthermore, there are other substantial considerations that argue for combining the low-phytate approach with the “high mineral” approach. Consider the cases of zinc and iron. In the case of zinc, the negative impact of dietary phytic acid is both on the zinc consumed in a meal, but also on endogenous zinc encountered in the intestinal tract, and this latter effect may contribute substantially to net zinc loss [57]. Thus in the case of populations that consume substantial amounts of phytic acid in the grain- and legume-based diets, the full benefits of elevated dietary zinc may be reduced by high phytate levels. 

Consider the results of a study that evaluated zinc nutrition with chicks consuming feeds prepared with a normal-phytate barley and barley *lpa*-M955, in which seed phytic acid is reduced 95% (Figure 5) [58]. Chicks were fed six different diets. These were prepared such that the sole source of phytic acid was either a wild-type (for grain phytic acid) barley near-isogenic line or a sibling near-isogenic line (“M955” for *Hvlpa*-M955) in which grain phytic acid is reduced by >90%. Further, diets prepared with each barley were supplemented with either 0.0, 10 mg/kg or 20 mg/kg zinc. The two grain types had similar endogenous zinc levels (about 23 to 24 mg Zn/kg). The results (Figure 5) clearly illustrate that zinc nutritional health, as measured by tibia zinc, was enhanced by zinc supplementation only in those animals consuming wild-type barley. Zinc nutritional health as assayed by tibia zinc was optimal in animals consuming the M955 barley regardless of zinc supplementation level. Thus, in the case of this low-phytate barley, increasing levels of zinc supplementation resulted in no added benefit. This clearly illustrates that the endogenous zinc levels of barley grain are adequate for optimal zinc health, if not for the endogenous phytic acid. From this perspective, zinc supplementation functions only to overcome reduced bioavailability due to dietary phytic acid. Should not this be a consideration in policy-making for approaches to deal with world-wide zinc deficiency?

Still, these results (Figure 5) do illustrate the value of zinc supplementation or biofortification for those crops/foods where seed-derived dietary phytic acid is at the high levels typical of many cereal- and legume-based foods. A good example of this is a recent evaluation of fractional and total zinc absorption in young rural Zambian children from meals prepared with zinc-biofortified maize [59]. Despite relatively high phytate levels, consumption of meals prepared with zinc-biofortified maize (34 μg Zn/g grain) resulted in a near-doubling of total absorbed zinc (1.1 mg Zn absorbed/day) when compared with that resulting from consumption of meals prepared with standard maize (21 μg Zn/g grain; 0.6 mg Zn absorbed/d). The authors note that this increase in total zinc absorption from the biofortified grain “occurred despite the high phytate concentration and high dietary phytate:zinc molar ratio of the biofortified grain” [59]. Of course one might wonder how combining “high grain zinc” with low-phytate might have further enhanced zinc bioavailability? 

In the case of iron, consider that as part of the Harvest Plus program (https://www.harvestplus.org/), common bean was chosen as one of the main targets for iron biofortification efforts. But in fact, the negative impact of phytic acid in foods prepared from a high-iron common bean produced via biofortification breeding led leading experts in the field to the conclusion that the common bean has “limited potential as a vehicle for iron biofortification” [60]. These stable isotope studies with Rwandese women found that any benefit in the consumption of meals prepared with iron-biofortified beans versus “normal iron” beans was negated by the phytic acid content of the bean. In a subsequent study, this same group found that fractional iron absorption was increased 60% to 130% (depending on polyphenol content) and total iron absorption was increased 60% to 163% (again depending on the polyphenol content) when foods were prepared using an *lpa* bean with a 90% phytic acid reduction as compared with a normal-phytate bean [60]. This group subsequently concluded [61] that “due to the low bioavailability of bean iron…exclusively breeding for high iron concentration may not provide enough additional absorbable iron to impact iron status” and that “the focus should now be on PA reduction”. Of relevance to questions of agronomic performance, the low-phytate bean line used in this study had previously been found to have little effect on plant performance or yield [62]. I will return to this last point below.

This group’s results and recommendations are informative when addressing what I view as the “paradigm-lock” in the biofortification field. The paradigm in question, as repeatedly stated by proponents of the biofortification approach [9,53,63], can be summarized as follows: (1) breeding for enhanced mineral density is feasible for many different staple food crops and is an “appropriate technology” for addressing micronutrient deficiency in the most at-risk populations in the developing world; (2) it will have little or no negative effects on crop yields but rather probably will enhance crop yields; (3) breeding for low-phytate may be problematic since dietary phytic acid may also have positive nutritional benefits that might be lost; (4) those who attempt breeding for low-phytate should proceed with caution since phytic acid is so important to a plant’s basic biology that breeding for low-phytate will inevitably harm plant growth and productivity. One can get the impression that in the case of breeding for enhanced mineral nutritional quality, biofortification breeding for enhanced mineral density is all good and low-phytate is all bad! 

I have a few problems with this paradigm. Take the case of zinc. Leading experts in the international nutrition field have identified phytate “as the most important inhibitor of zinc absorption in adult human diets” [64]. The model and equation currently used for determining the bioavailable zinc in food [65] requires knowledge of the phytic acid content of that food. The European Food Safety Authority (EFSA) used this model to generate a new set of dietary zinc recommendations for adults based on four levels of dietary phytate [66]. Seed-derived dietary phytic acid plays a similar negative role in iron bioavailability and deficiency [15,16,47]. Therefore it seems obvious that addressing the phytic acid content of foods is critical to the question of iron and zinc deficiency. In this regard, it is telling that in two recent reviews of progress in iron and zinc biofortification [54,67] there is not one mention of phytic acid or phytic acid:zinc molar ratios. In these reviews, curiously, there is also no mention of negative or inconclusive results, such as those reported by Petry et al. [55]. 

But in cases such as foods prepared with legumes or whole grains, breeding for enhanced mineral density may not be needed at all, if one breeds for reduced phytic acid. As indicated in the studies with barley (zinc) and beans (iron) discussed above [58,60], simply reducing phytic acid may provide sufficient enhancement of both zinc and iron bioavailability in many foods. Furthermore, consider the results of analysis of iron bio-availability from grain of maize wild-type and *lpa*1-1 lines, the later producing grain with a ~66% reduction in phytic acid, when assayed using the human Caco-2 (colon adenocarcinoma) cell in vitro assay (Figure 6; unpublished results kindly provided by Ray Glahn, USDA-ARS). The Caco-2 assay was designed to study the intestinal absorption of nutrients [68,69]. Use of this assay indicated that the addition of ascorbic acid, a known enhancer of iron absorption, to wild-type maize flour more than doubled iron uptake observed in assays of wild-type maize with no ascorbic acid (Figure 6). Importantly, iron uptake from *lpa*1-1 maize flour was essentially identical to that observed from wild-type flour plus ascorbic acid. Furthermore, the addition of ascorbic acid to *lpa*1-1 maize flour further increased iron uptake by 40%. 

These Caco-2 results provide further evidence that the inherent nutritional value of *lpa* grain can provide benefits equal to those achieved via supplementation or via the use of additives like ascorbic acid. This view is supported by the results of a recent analysis of the iron content and bioavailability, as determined via the Caco-2 assay, of the maize germ versus endosperm [70]. In maize, the germ fraction contains ~80% of the grain’s total phytic acid, and 27–54% of its iron. Yet this iron is poorly bioavailable. Removal of the germ enhanced the iron bioavailability of the remaining grain fraction (endosperm plus aleurone), indeed to the extent that there was more bioavailable iron, as determined by the Caco-2 assay, in de-germed maize flour than in common bean flour, despite the fact that the common bean flower contained five times the absolute level of iron. Phytic acid is clearly the culprit here. 

Most biofortification breeding efforts and nutritional studies of biofortified foods only address one micronutrient, iron or zinc or calcium for example. That simply reflects the standard approach to these types of programs and studies. But in viewing the real world of “global” mineral nutrition, what about the fact that greatly reducing dietary phytic acid will probably enhance bioavailability of all of these? Of course, there are many cases such as foods obtained following milling of grain crops where following the removal of bran the resulting food is a relatively poor source of iron or zinc yet is also low in phytic acid. In those cases, such as white rice, breeding for enhanced mineral density has many straightforward beneficial applications where the low-phytate trait might have little benefit.

The *lpa* beans used in the studies of Petry et al. [60,61] were not isolated with support from Harvest Plus [62]. In fact to the best of my knowledge, to date Harvest Plus and international agricultural centers participating in Gates Foundation-supported biofortification efforts have not provided support for the development of any low-phytate crops. Has this lack of support for “the *lpa* approach” by the most prominent programs in crop nutritional quality breeding for the international community been to the benefit or detriment of those populations at greatest risk for mineral deficiency? Consider the truly global improvement in mineral nutritional health obtained via the consumption of *lpa* types documented above. When field-dominating programs develop broad strategies that target specific approaches, a “canalization effect’ can occur: support for other approaches might be negatively impacted. Consider the progress that might have been made in the breeding of low-phytate crops if it had the support of such well-funded programs, not post 2015 following the results of Petry et al. [60], but about 1995 about when low-phytate genetic variants first became available? 

Biofortification breeding for mineral density may have another problem: it ignores the fact that in many societies, grain and legume crops or products made from them are used both in human foods and animal feeds. Thus an *lpa* crop might be more broadly useful and simply more practical. Consider the cases of rice. The vast majority of rice is consumed as white rice consisting almost entirely of central endosperm, the product of milling/polishing process that removes the germ and aleurone layer as the “bran fraction”. Since in the cereal grains nearly all the phytic acid and most mineral stores are found in the germ and aleurone layer, they end up in the bran fraction. Rice bran is an important and valuable side-product in areas of major rice production and is used in animal feeds [71]. As a result, while a low-phytate mutation that primarily impacts the chemistry of the bran fraction might be relatively less valuable for white rice improvement, low-phytate bran would be far more nutritious than a “normal-phytate” bran when used in animal feeds. When chickens were fed a diet where rice bran, wheat bran, corn bran, soy bran, and oat hulls were used as fiber, only the ones fed rice bran had reduced body growth [71]. This was attributed to a higher phytic acid level in the rice bran diet (1.3%) as compared with that in other diets (0–0.4%). Furthermore, where rice is consumed as a whole-grain “brown rice”, or for that matter any cereal grain crop when consumed as a whole-grain, the low-phytate trait would be of value for both human and animal consumption. Therefore, policy and research direction should more often take into account the value of *lpa* types both in human and animal feeds. 

The quality of judgment about questions pertaining to the agronomics of breeding for enhanced mineral density versus low-phytate may have also been less than optimal. For example, the claim that breeding for enhanced mineral density will have little or no effect on yield but rather will probably enhance yield is not uniformly supported by published results. A thorough and well-conducted study of the relationship between grain yield and grain zinc concentration of zinc-biofortified rice cultivars, conducted over four locations and five years, found “high grain Zn concentration was at the cost of grain yield” [72]. Of course, a negative relationship between high seed mineral density and yield will not always be the case, but statements that imply that such a negative relationship will not likely be the case are misleading for several reasons [56]. For example, it is well known that there is often an inverse relationship between grain yield and grain mineral density [56]. There is a straightforward biological explanation for this: as “harvest index” (the amount of grain produced by a unit of crop biomass) increases, the concentration of minerals per unit of grain decreases. Thus the inverse of this suggests that one of the easiest ways to select for enhanced seed mineral density would be to select for reduced yield! The well-known negative relationship between grain yield and grain mineral density may explain the observation that enhanced iron and zinc density resulting from biofortification breeding is sometimes accompanied by increased phytic acid [55,59,73]. Of course, this could be due to other factors such as selection for transport functions that enhance the concentration of all three minerals simultaneously. But given the published observations, should not one ask if breeding for enhanced gain iron or zinc density typically results in elevated phytate levels?

With all of that said, an important recent study found that the adoption of improved “iron biofortified beans” by smallholder farmers in Rwanda led to a 23% increase in yields and a “potential” 24% increase in farmer income [74]. While non-breeding or non-biological factors not related to enhanced grain mineral density might have contributed to these benefits, such as the simple introduction of advanced cultivars or other cultural or economic factors, this represents a notable outcome and will factor into future policy supporting the adoption of biofortified crops. However, the assumption in the above “biofortification paradigm” that *lpa* types will inevitably have poor field performance, is also not supported by all agronomic studies or by crop breeding theory (see below).

What was the rationale behind this lack of support for the “low-phytate approach”, given its potential? Perhaps it simply reflects a justifiably conservative approach to crop improvement strategies for those crops providing staple foods for at-risk populations in the developing world. After all, there may be some fundamental flaws for the low-phytate trait. For example, perhaps the change in seed P chemistry that results in reduced phytic acid P accompanied by elevated inorganic P might increase the incidence of fungal infection, which could increase the incidence of mycotoxin contamination [75]. Alternatively, the rationale might reflect a concern that dietary phytic acid also may have health-beneficial roles, as an anti-oxidant and anti-cancer agent or as an inhibitor of renal stone formation (reviewed in [76]). While highly significant, these are largely problems of aging in the developed world, but the vast majority of persons in mineral deficiency at-risk populations are primarily infants, children and women of child-bearing age of lower economic status in the developing world. For these individuals, malnutrition leading to “stunting” in adolescents, or iron-deficiency anemia in women, is of more pressing concern than is cancer or kidney stones. 

But there is a second issue with the claim of health benefits for dietary phytic acid; some of the claims may be problematic. While the anti-oxidant and mineral chelation role of dietary phytic acid in the digestive tract is based on good science, claims for health benefits that require uptake and transport in bodily fluids such as blood or urine, for example, the claimed role in preventing renal stone formation, probably are unfounded, since a recent study demonstrated that there is no phytate in human plasma or urine [77]. These authors, in fact, conclude that administering phytate as a dietary supplement “in the hope that it might impact directly or indirectly on cancer or other pathologies seems highly inadvisable” [77]. The problem of widespread transmission of this false claim is not helped by a research article with the deceptive title “[3H] Phytic acid (inositol hexaphosphate) is absorbed and distributed to various tissues in rats” [78]. If one reads that paper, one finds that it was not phytic acid but primarily it’s inositol backbone that was absorbed and distributed. myo-Inositol can have health beneficial effects, but myo-inositol is not phytic acid. Thus dietary phytate no doubt has some beneficial roles, but they probably are much fewer then is often claimed, including claims cited in papers advocating for biofortification breeding of high mineral density.

## 4. The Low-Phytate Trait, Yield and Seed/Grain Quality: Issues and Attitudes 

### 4.1. Does the Low-Phytate Trait Mean Low Yield or Poor Seed Quality? Yes, If You Only Have Six Months to Breed or Engineer It! 

One would be hard-pressed to find research reports or reviews of the low-phytate trait that do not begin with the statement to the effect that while it has many benefits, it is obligately associated with negative effects on plant and seed performance and yield. But as time goes there have been some reports of low-phytate genotypes of various crops, whether developed via conventional breeding or via genetic engineering, which have good field performance and yield [27,32,62,79]. Perhaps this attitude might slowly change as these developments become more widely known [80].

To be clear, studies indicate that alleles of genes that condition the *lpa* trait, including variant alleles of the same gene, can have greatly differing impacts on plant or seed performance. An excellent example of this is the maize *lpa*1 gene, alleles of which range from having a modest impact on plant and seed performance and yield [81] to lethality [82]. Maize *lpa*1 encodes one of the maize genome’s multiple multidrug resistance-associated protein (MRP) genes, which encodes an ABC transporter specific to phytic acid transport [33]. Of the first 20 or so alleles of maize *lpa*1 that we isolated, two were lethal as homozygotes [82]. Furthermore, studies of another allele of maize *lpa*1, *lpa*1-241, found it negatively impacted seed viability via reduced protection against oxidative stress during seed maturation and storage, indicating a protective role for phytic acid as an anti-oxidant [83]. In contrast, recent studies of a rice *lpa* mutant (perturbed gene unknown), 9311-*lpa*, indicated that it’s delayed or impaired germination, as compared with its non-mutant parental line, may be due to decreased ROS (reactive oxygen species ) in seed tissues during germination [84]. This, in turn, is possibly due to a higher level of InsP_3_ rather than to detoxification of the ROS burst during germination. 

Alleles of an MRP gene provides another instructive example via studies of the low-phytate soybean and common bean lines conditioned by recessive alleles of their genomes’ MRP genes. Each of these two close relatives’ genomes contains multiple copies of this gene, two in the common bean and three in the soybean. The soybean low-phytate line CX1834 line [85], homozygous for the first soybean *lpa* mutations, was found to be conditioned by mutations in two copies of this particular MRP gene [86]. Initial studies of its agronomic properties revealed the interesting “seed source effect”: via a still unknown mechanism, seed maturation of this genotype in a tropical environment (Puerto Rico) resulted in negative effects on subsequent germination, emergence and yield not observed when seed maturation occurred in a temperate environment (Iowa) [87]. However, a common bean low-phytate mutant conditioned by a mutation in only one of the common bean’s two MRP duplications displays little or no negative impact on germination, emergence or subsequent plant growth, performance or yield [62,88]. Furthermore, a sustained breeding effort [79] using the soybean CX1834 line as the source of the low-phytate trait produced two soybean lines whose yields and germination were statistically similar to high-yielding check cultivars (discussed below). These studies of maize and bean mutations in MRP genes, and studies of other *lpa* mutants [84], also illustrate that apart from any agronomic value, *lpa* genetics represent a valuable resource in studies of plant and seed biology and the role that the phytic acid and inositol phosphate pathways may play in it. 

But the main point is that yes, one can readily isolate *lpa* alleles homozygosity for which greatly impact performance and yield. If a program isolates *lpa* mutations in a given crop, or engineers a knock-out of a phytic acid pathway gene, and simply evaluates performance of early-generation homozygotes, the chance that they will observe poor performance is fairly high. For example, if nine out of ten randomly-isolated alleles have a clear negative impact, as was the case with maize *lpa*1, then the chance of reporting an *lpa* line with poor yield is pretty high. I think this is exactly what has happened. But with hard work and some breeding, that one-in-ten high-yielding *lpa* line can probably be developed. 

Very good examples of this are field trials of barley *lpa* lines, representing variant alleles of at least four genes (Figure 7, [27,89]). Four pairs of near-isogenic lines were included in both rounds of trials and those will be discussed here. These pairs consisted of a wild-type sibling and a sibling homozygous for one of four independent *lpa* mutations: barley *lpa*1-1 conditioning a ~47% reduction in seed phytic acid; barley *lpa*2-1 conditioning a 50% reduction in seed phytic acid; barley *lpa*3-1 conditioning a ~65% reduction in seed phytic acid; barley *lpa*-M955, conditioning a ~95% reduction in seed phytic acid (also illustrated in part in Figure 4). The appropriate comparison in these studies is between the wild-type (“normal phytic acid”) sib line versus the homozygous *lpa* sib line in each pair of near-isogenic lines. For the first round of trials in 2002 and 2003, these homozygous mutant or wild-type lines were obtained after only two to four generations of backcrossing to the recurrent parent, the progenitor cultivar “Harrington”, followed by self-pollination. Lines for subsequent study were selected via visual inspection for plants displaying nice growth equivalent to the wild-type progenitor cultivar “Harrington”. However, no rigorous selection for plant performance or seed weight or quality within the *lpa* class in a given generation of backcrossing was conducted; in each generation seed from essentially all members of a given progeny class (wild-type or mutant) were bulked to plant the next generation.

This first round of field trials were conducted in 2002 and 2003 in four locations in Idaho USA, two of which were irrigated and two of which were only “rain-fed”, the latter representing much more stressful and less productive environments (Figure 7A,B). The first important observation is that in the irrigated trials in 2002–2003, we observed no effect of homozygosity for *lpa*1-1 on yield (Figure 7A, left). However, with larger decreases in grain phytic conditioned by *lpa*2-1, *lpa*3-1, and *lpa*-M955 respectively, a very linear and negative relationship between phytic acid reduction and yield was observed (indicated by the red regression line). The second important observation in the 2002-2003 trials is that the absent or modest reductions in yield in the *lpa* sib lines for the *lpa*1-1, *lpa*2-1, and *lpa*3-1 mutations when assayed in irrigated trials is much more pronounced in the rain-fed trials (Figure 7B). Although not statistically significant in every case, there is clearly a much larger impact of the *lpa* trait on yield, including for the *lpa*1-1 sib line, in this first round of trials in the rain-fed versus irrigated locations. Thus in this first round of field trials: (1) the *lpa*1 line performed best overall and its performance in irrigated locations was excellent; (2) there appeared to be a very linear and negative relationship between the level of phytate reduction and yield in the irrigated trials; (3) this negative impact on yield was very pronounced in the more stressful, non-irrigated locations.

Breeding with these *lpa* mutations continued and a second round of field trials were conducted in 2010 and 2011 (Figure 7C,D; [27]). For this second round seed was obtained following further backcrossing (four or five generations) but again, no rigorous selection for plant performance or seed traits within a given *lpa* progeny class was conducted. When subsequently tested over two years in five locations in Idaho, USA, including locations that again were either irrigated or not, barley *lpa*1-1 had no discernable effect on yield, regardless of location. Furthermore, in the irrigated locations (Figure 7C), the difference in performance between a given wild-type sib line versus its homozygous *lpa* sib was much less pronounced than in 2002 and 2003: yield of *lpa*1-1, *lpa*2-1 and *lpa*3-1 homozygous mutant lines were as good or better than their wild-type sib lines and the loss of yield conditioned by *lpa*-M955, as compared with its wild-type sib, was less dramatic than that observed in 2002 and 2003. While there still was a clear, negative relationship between yield and phytic acid reduction (indicated by the red regression line), the impact of phytic acid reduction on yield was approximately half of that observed in 2002: the regression coefficient was -0.035 in 2002/2003 (Figure 7A) versus -0.018 in 2010/2011 (Figure 7C). One cannot rule out the possibility that this difference in regression of yield against phytic acid reduction is due in whole or part to differences in overall production conditions between 2002/2003 and 2010/2011. For example, overall, yields were lower in the irrigated locations in 2010/2011 as compared with 2002/2003. 

While field performance of the barley *lpa* lines was again much more impacted in the far more stressful rain-fed locations (Figure 7D), in 2010 and 2011 *lpa*1-1 still yielded as good as its wild-type sib. The excellent field performance of barley *lpa*1-1 probably is due to the fact that this mutation’s phenotype is highly tissue-specific, reducing phytic acid accumulation in the aleurone but not the germ, the two sites of phytic acid accumulation in cereal grains [26,27]. Since the germ of barley *lpa*1-1 is wild-type in terms of phytic acid, the impact of this mutation on germination, emergence, plant growth, and performance is minimal.

Taken together, these data indicate that even a minimal amount of breeding, limited to backcrossing, works! This conclusion is supported by the results of backcrossing the first soybean low-phytate mutations [85] into an elite background in a breeding program at the Univ. or Tennessee (USA, [79]). Two low-phytate soybean lines were obtained after five generations of backcrossing to an elite line. Field trials of these lines found their yield statistically equivalent to the yield of two high-yielding culivars included as checks, and also found no effect of the low-phytate trait on germination. Thus good germination, field performance and yield of low-phytate types has now been documented in two important food legumes, the common bean and the soybean [62,79]. 

However, what is lacking in these as well as in all other cases of breeding with *lpa* genotypes is recurrent selection within the *lpa* progeny class, over several generations, for seed and plant performance traits. For example, while studies show that recurrent selection might improve the germination and emergence of low-phytate soybean lines [90], the results of such selection have not been reported to date. 

There are two additional problems with the viewpoint that the *lpa* trait is obligately associated with reduced yield or seed quality that cannot be overcome with traditional crop breeding methods. First, the genetics of the *lpa* trait is still in its infancy, as compared with other seed chemistry traits like starch, protein or oil content or chemistry. In addition to the relatively few major *lpa* loci that have been identified to date, to my knowledge there has been only two studies to date documenting secondary loci or allelic variants of genes that have a valuable modifier effect in an *lpa* background (discussed below, [91,92]). Such modifiers are well known for genes that perturb or alter starch, protein and oil content and chemistry, and critically important to breeding elite-performance lines with such traits [93,94,95]. Consider the genes and loci that modify starch, sugar or carbohydrate content in maize. Not only are already well-known modifiers important in breeding and end-use quality, such as sugary enhancer used to modify *sugary* in sweet corn breeding [93], but additional loci continue to be identified that have modifying effects. These new discoveries continue even after many decades of intensive research, thus the “supply” has not been exhausted, even in the case of truly major seed constituents such as starch, oil and protein. 

The first genetic engineering of a gene to serve as a modifier of poor fertility and growth of an *lpa* mutant targeted the *Arabidopsis “*Gle mRNA export factor” [91]. The negative effects on vegetative growth and fertility observed in low-seed phytate *Arabidopsis* lines conditioned by knock-out of its *Ipk1* (inositol pentak*is*phosphate 2-kinase) gene were rescued by engineering allelic variants of the Gle mRNA export factor for elevated Ins P_6_ sensitivity [91]. 

A sustained classical breeding effort can also overcome seed quality issues associated with the low-phytate trait via selection for beneficial modifiers. An excellent example of this concerns efforts to combine the low-phytate and low-saturated fat traits in soybean. Low-saturated fat represents an important health-beneficial trait for soybean oil. The U.S. Food and Drug Administration set an upper threshold of 89 g saturated fats per kg product in order to refer to a product as “low saturated fat” [96]. A breeding program targeted developing low-phyate/low saturated fat soybean cultivars [92,97]. The low-phytate donor line was the original low-phytate soybean mutant CX1834 [85], and its seed had a relatively high saturated fatty acid (palmitate + stearate) concentration ranging from ~100 to 174 g per kg seed [97]. This line was crossed with a “low saturate donor parent”, a line homozygous for alleles of two genes that confer the low-saturated fat seed phenotype, and whose seed saturated fatty acid concentration was 71 g per kg seed [97]. Normal-phytate and low-phytate progeny classes were identified following one generation of backcrossing to the low-saturate parent (followed by self pollination), and amongst these two classes, 20 normal-phytate and 20 low-phytate progenies were further identified that produced seed with less than 50 g palmitate per kg. These lines were then field tested and the normal-phyate and low-phytate progenies were found to produce seed that had 77 and 83 g palmitate+stearate per kg seed, respectively. This difference was not statistically significant and both levels are below the FDA’s threshold, but the level of saturated fatty acids in seed of the low-phytate lines was still consistently higher than that observed in the normal-phytate progenies and higher than that observed in the standard low-saturated fat parent. These lines produced seed that was deemed to close to the FDA threshold to warrant recommendation for commercial production [97]. 

How this group’s work illustrates the value of a sustained effort and identification of beneficial modifier alleles is that following further crossing and selection, low-phyate soybean lines with saturated fatty acid levels less than 70 g per kg seed were successfully identified. It was concluded that this was accomplished via selection for “favorable modifying genes for low saturate concentration” [92].

A problem associated with rice *lpa*1 is that it is associated with grain “chalkiness”, an undesirable characteristic [98]. This is reminiscent of the case of opaque *2*/high-lysine maize. Perhaps the most important and relevant example of the problem of field performance for a nutritionally-enhanced major crop is the case of opaque *2*/high-lysine maize. If anyone well versed in the history of efforts to breed nutritionally-enhanced staple crops read that there is concern with the agronomic performance of the early generations of *lpa* crops, that person might think of the story of opaque2/high-lysine corn [94]. After its initial identification, excitement led to disappointment; homozygosity for *o*2 resulted in undesirable grain characteristics including chalkiness, and impaired field performance and yield. But addressing the amino acid deficiencies such as lysine deficiency of standard maize is critically important to nations like Mexico where maize is such an important staple food. As a result, there has been substantial government and institutional support for this work in several nations. Sustained breeding and research over several decades has yielded high-performance high-lysine, “Quality Protein” maize with excellent yields and good grain characteristics. Elite performance of any line or hybrid, regardless of specific end-use or quality traits they might contain, reflects favorable combinations of alleles at large numbers of dispersed loci that only result from sustained and ongoing breeding over many generations. As discussed above, reports of recurrent selection for performance within a *lpa* line, or in fact any sustained effort at *lpa* breeding, has been very limited. In a parallel to the *o2* story, perhaps recurrent selection might correct the problem of “chalkiness” in rice *lpa*1. 

Second, recurrent selection within a *lpa* line for field performance or yield probably would not only select for favorable genetic alleles, in the sense of sequence-differing variants, but also will very likely select for favorable epigenetic variants [99]. This latter type of variation, and its role in breeding crop lines with elite performance, is not yet well understood in terms of science nor is its importance appreciated in the crop breeding community. I predict that that will change greatly over the next decade.

This discussion so far has only addressed “conventional” approaches to breeding high-yielding low-phytate crops. Clearly, genetic engineering for the trait provides an additional and powerful approach to developing low-phytate crops with good performance characteristics [33]. There are a growing number of targets for such engineering [3,4,5,31,32,33,36,91]. It is interesting that in maize, one of the worlds’ most important food and feed crops, both the first use of zinc-finger nucleases and more recently the methods of TALENS and CRISPR/Cas for sequence-targeted gene engineering, targeted genes involved in the low-phytate trait [100,101]. Perhaps due to the significant potential benefits of the trait, developing high-yielding stress-tolerant low-phytate crops has become something of a holy grail for crop genetic engineering. Perhaps the most powerful approach might turn out to be engineering crops with seed tissue-targeted phytase overexpression. This has been accomplished with a growing number of crops including maize [102], soybean [103] and barley [104]. This approach can result in quantitative conversion of phytate P to inorganic P in mature seed [103], but would also provide active phytase action against any source of phytate in a food or feed. 

### 4.2. Is Comparing the Yield of a Low-Phytate Type Against a Wild-Type Like “Comparing Apples and Oranges”, or More Appropriately, Comparing Yields of Sweet Corn with Field (Starchy) Corn?

One would not insist that sweet corn yields compare favorably with field or starchy corn yields. But is not that exactly what people have been doing with the low-phytate trait? The problem is that in the context of crop production for mainstream poultry and swine production in the U.S., often *lpa* crops must compete with standard commodity crops. In the industry’s primarily horizontal structure, *lpa* crops have to yield close to standard crops and would also have to be handled as a “specialty-use crop”, where they are segregated in both storage and shipment. In light of the fact that a practical alternative to the problem that *lpa* crops address in this context already exists, the feed additive phytase, the “yield issue” and need for segregated production and handling present barriers to widespread production. But should *lpa* types be compared directly with standard types? For example, in the case of maize, should not the *lpa* type simply be considered a “specialty use” corn or perhaps represent a new maize commodity classification? For example, in the case of maize, we have “dent corn”, “flint corn”, “sweet corn”, “waxy corn”, and “amylomaize”. To this we should add low-phytate maize. Would it be appropriate to compare the yield of waxy corn, or sweet corn, with a starchy field corn? Absolutely not. In light of the numerous benefits of low-phytate types as human foods or animal feeds, should one not grow a nutritionally-enhanced crop variant that perhaps has 5% to 10% less yield than a standard variant but one that is substantially more nutritious? 

## 5. Conclusions

The *lpa* or “low-phytate” seed trait can provide numerous potential benefits to the nutritional quality of foods and feeds and to the sustainability of agricultural production. These include enhanced phosphorus management contributing to enhanced sustainability in non-ruminant (poultry, swine, and fish) production; reduced environmental impact via reduced waste P in non-ruminant production; enhanced “global” bioavailability of minerals (iron, zinc, calcium, magnesium) for both humans and non-ruminant animals; altered distribution of minerals in cereal grains potentially resulting in enhanced mineral contents of milled products; enhancement of animal health, productivity and the quality of animal products; potential enhancement of food and feed protein and starch utilization; development of “low seed total P” crops which also can enhance management of P in agricultural production and contribute to its sustainability. Evaluations of this trait by industry, and by advocates of biofortification via breeding for enhanced mineral density, have been too short term and too narrowly focused. Arguments against breeding for the low-phytate trait overstate the negatives such as potentially reduced yields and field performance or possible reductions in phytic acid’s health benefits. Progress in breeding or genetically-engineering high-yielding stress-tolerant low-phytate crops continues. While there are widely available and efficacious alternative approaches to deal with the problems posed by seed-derived dietary phytic acid, such as use of the enzyme phytase as a feed additive, or biofortification breeding, if there were an interest in developing low-phytate crops with good field performance or good seed-quality, it could be accomplished given adequate time and support. Perhaps due to the potential benefits of the low-phytate trait, the challenge of developing high-yielding, stress-tolerant low-phytate crops has become something of a holy grail for crop genetic engineering. Even with a moderate reduction in yield, in light of the numerous benefits of low-phytate types as human foods or animal feeds, should one not grow a nutritionally-enhanced crop variant that perhaps has 5% to 10% less yield than a standard variant but one that is substantially more nutritious? Such crops would be a benefit to human nutrition especially in populations at risk for iron and zinc deficiency, and a benefit to the sustainability of agricultural production.

## Figures and Tables

**Figure 1 plants-09-00140-f001:**
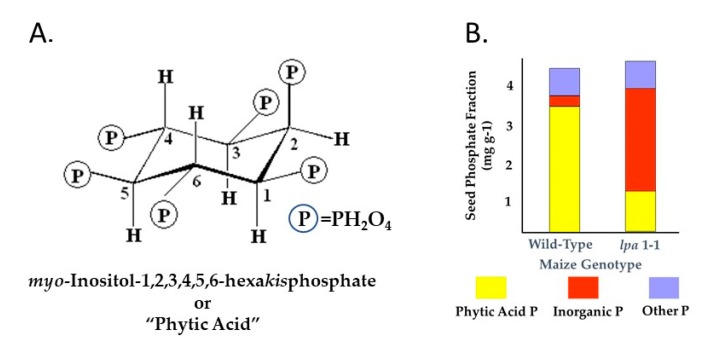
(**A**) Phytic acid, myo-inositol-1,2,3,4,5,6-hexakisphosphate. (**B**) The basic seed phosphorus (P) phenotype of “normal phytic acid” or wild-type and *low phytic acid* (*lpa*) genotypes, as illustrated by the initial analyses of wild-type and *low phytic acid* 1-1 maize isolines [2]. While seed total P may vary in any given genotype depending on various environmental, non-genetic factors, the relative contributions of phytic acid P, inorganic P and “other P” to seed total P typically remain fairly constant. “Other P” refers to all P other than phytic acid P and inorganic P, such as P found in other inositol phosphates, DNA, RNA, protein, carbohydrate, lipids, etc.

**Figure 2 plants-09-00140-f002:**
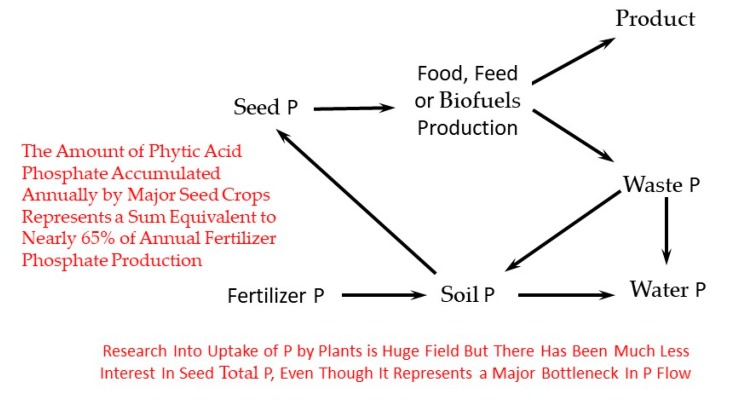
Phosphorus (P) flow through the world-wide agricultural ecosystem.

**Figure 3 plants-09-00140-f003:**
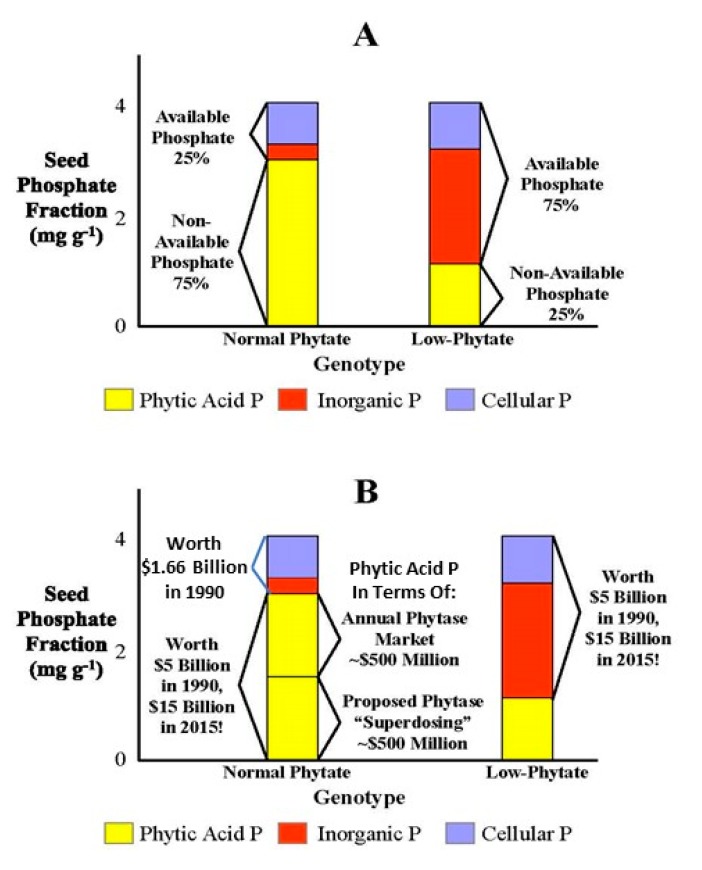
(**A**) Typical seed phosphorus (P) fractions and their bioavailability to non-ruminant animals, in a wild-type or “normal phytate” crop type versus a low-phytate line. “Available phosphate” and “non-available phosphate” refers to that fraction of seed total P that would be absorbed and utilized following consumption by non-ruminants. This is only an illustration for discussion purposes and does not represent actual data for any given crop or low-phytate line. Cereal crops usually have 3.0 to 4.0 mg Total P g^-1^, such as in the illustration. Legume and oil-seed crops can often have 5.0 or more mg Total P g^-1^, but the relative proportions of phytic acid P, inorganic P and “cellular P” remain similar to that observed in cereal grains. “Cellular P” refers to all P other than phytic acid P and inorganic P, such as P found in DNA, RNA, protein, carbohydrate, lipids, etc. (**B**) The same bar graph of seed P fractions but with: (1) the relative value in U.S. dollars of the respective seed P fractions, expressed in terms of the market price for rock phosphate (adjusted for inflation), in 1990 versus 2015 and (2) the relative value of the current “phytase market” and the proposed “superdosing” of phytase.

**Figure 4 plants-09-00140-f004:**
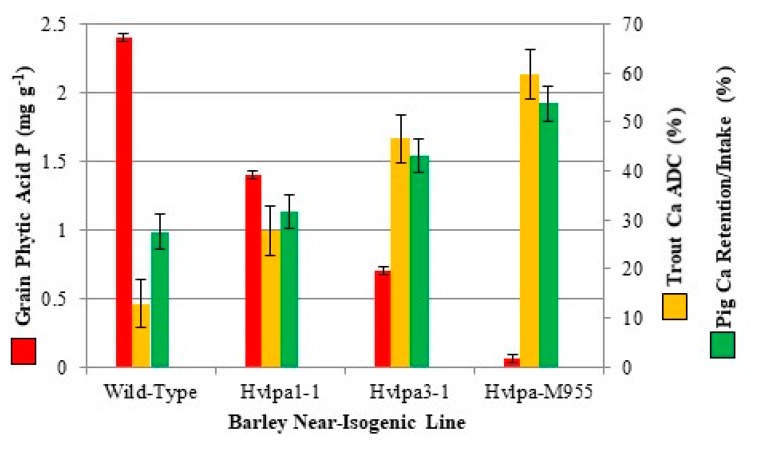
The highly linear negative relationship between dietary calcium bioavailability and grain phytic acid level, as observed in two separate studies, one with pigs [44] and one with trout [45]. Calcium bioavailability was measured in trout as “apparent digestibility coefficient” (ADC) and in pigs as “% retention/intake”. Animals were fed diets prepared with four barley near-isogenic lines produced in the same location: (1) wild-type (for grain phytic acid, the cv. Harrington); *Hvlpa*1-1 with a ~40% reduction in grain phytic acid; *Hvlpa*3-1 with a ~70% reduction in grain phytic acid; and *Hvlpa*-M955 with a >95% reduction in grain phytic acid. Error bars are the standard deviation of the mean or “standard error” (SE) for each variable: grain phytic acid P, SE = 0.03, n = 3; Trout Ca ADC %, SE = 5.01, n = 2; Pig Ca retention/intake %, SE = 3.50, n = 5.

**Figure 5 plants-09-00140-f005:**
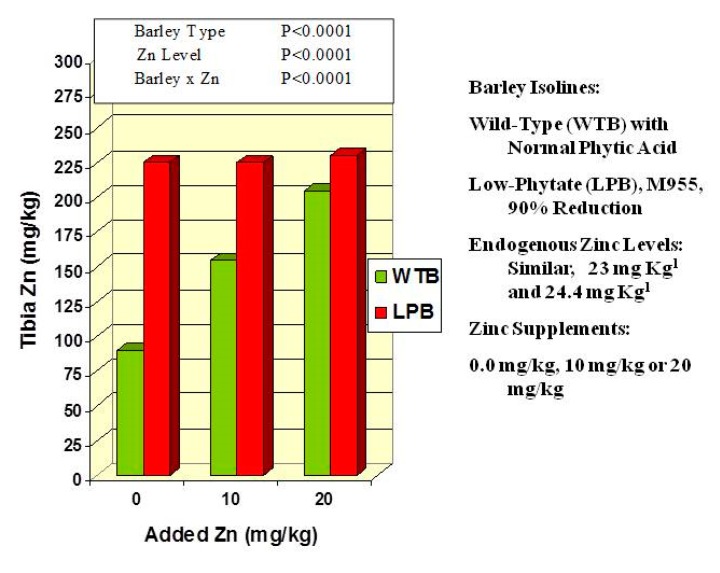
Effect of barley grain phytic acid level and supplemental zinc on chick tibia zinc at 21 days of age. The standard deviation of the mean or “standard error” (“SE”) was 6.08, n = 4.

**Figure 6 plants-09-00140-f006:**
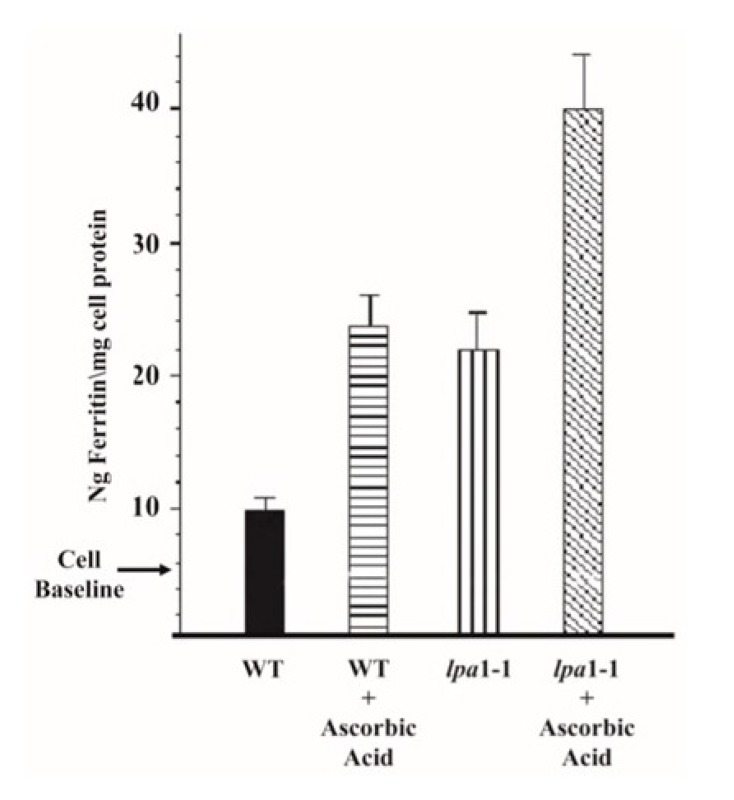
Caco-2 Assay for iron bioavailability from flour produced from maize wild-type (WT) and *lpa*1-1 isohybrid grain. The *lpa*1-1 grain contained 66% less phytic acid than WT. Bars indicated standard deviations, n = 3 (data provided by Ray Glahn, USDA-ARS Plant, Soil and Nutrition Lab, Cornell University, Ithaca, NY, USA).

**Figure 7 plants-09-00140-f007:**
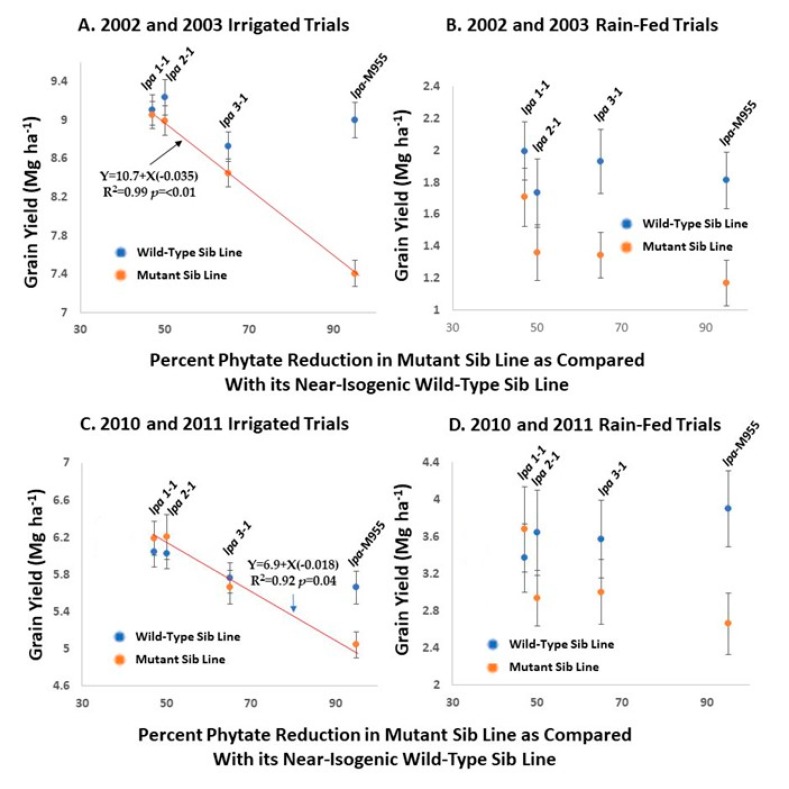
Two rounds of yield trials of barley *low phytic acid* (*lpa*) near-isogenic wild-type (normal phytic acid) lines, conducted in 2002 and 2003 and then in 2010 and 2011. Four pairs of near-isogenic lines, each pair consisting of a wild-type (normal phytic acid) sib and a *lpa* sib, were evaluated in several locations in Idaho (USA). These locations were either irrigated (two locations in 2002/2003 and three locations in 2010/2011) or rain-fed (non-irrigated; two locations in both 2002/2003 and 2010/2011) [27,89]. The homozygous *lpa* member of each isoline pair are: *lpa*1-1, which conditions a ~47% reduction in seed phytic acid; *lpa*2-1, which conditions a ~50% in seed phytic acid; *lpa*3-1, which conditions a ~65% reduction in seed phytic acid; *lpa*-M955, which conditions a ~95% reduction in seed phytic acid. (**A**,**B**): Trials conducted in 2002 and 2003. (**C**,**D**): Trials conducted in 2010 and 2011. The error bars represent the standard deviation of the mean, or standard error, n = 12 to n = 36. Linear regression of yield against % phytate reduction (indicated by red lines) was calculated for the *lpa* isolines in the irrigated trials.

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
