# Peer review of "Low phytic acid Crops: Observations Based on Four Decades of Research"

_plants, 2020, doi:10.3390/plants9020140_

Round 1

Reviewer 1 Report

The submitted paper corresponds to a position paper of the author on the development of low phytic acid plants more than a real review. It gives interesting data and summarized the attempts to breed low phytic acid producing plants and overviews positive and negative arguments for their development and use.

However, the paper appears more militant than objective. It does not give any numerical data concerning the positive arguments for example to reduce P wastes or to increase mineral availability, nor calculates benefits from numeric simulation models, except for the economy due to the increasing price of P input.

He defends the fact that both benefits for animals and humans must be addressed to evaluate the benefits of lpa mutants but does not develop a real multi-criteria analysis. For example, it is said that the reduced plant yield observed for lpa mutant can be reduced if irrigation is applied but no real economic evaluation was made to see if this type of mutants is still interesting taking into account this fact, especially in a context of climate changes and necessary increase of sustainability.

The different arguments and discussion are not very efficiently organized.

There is no sufficient evaluation taking into account that it exists a relationship between P/phytic acid reduction and mineral concentration (Zhao FJ et al., Journal of Cereal Science 49 (2009) 290–295).

The author also does not sufficiently discussed the fact that problem of phytic acid presence is different depending on the grain nature (cereal  or pulse) and the process used for its transformation.

The author does not sufficiently discussed the fact that lpa mutant also display a delay in germination as recently shown (Zhou L. et al. Plant Growth Regulation (2018) 85:411–424).

More problematic is the assumption that because only inositol is absorbed and distributed to tissues it is not phytate that is active for health effect mainly as an antioxidant and preventing molecule against cancer. But, degradation of phytate results in myoinositol therefore it appears to be potentially involved in these properties.

Figure 1 is not easy to understand and must be improved.

Generally there are no sufficient number of figures to illustrate data and ideas that are discussed.

Author Response

I would like to thank Reviewer 1 for the constructive and helpful comments and suggestions. I have endeavored to incorporate or address all the suggestions. 

1) This paper was not intended as a "real review", nor a thorough review (I have already written both many times), nor as a position paper. Rather it is simply a commentary reflecting views developed over four decades of work. Views are stated but careful reading would reveal that in every case, the alternatives are noted and given full credit. In any case, the paper was rewritten to be less "self-referential" and to sound less "militant", but the perception of it being "militant" cannot be totally avoided. When one critisizes broadly held views taken by others and such views dominating the field for literally decades, that probably will end up sounding militant. But such well-reasoned and even-handed criticism has its place.

2) I am not sure I understand the Reviewer's second comment. I clearly demonstrate that good yields in both irrigated and non-irrigated or stressful production environments have been achieved for certain low-phytate types. The point is that reduced plant yield is not obligatory for low-phytate crops. I'm not sure what a "real multi-criteria analysis" is, but such an analysis was not intended. I think the main points are actually very clear and staightforward. I challenge anyone who looks at reports having to do with the benefits of biofortification for humans to find one that discusses the roles of phytic acid asa  key player in both mineral nutrition in humans and P nutrition in animal agriculture. That is a main point and additional analyses are not needed to make it. 

3) I agree with the reviewer that the paper needed better organization and I have reworked the paper to improve organization. 

4) In low-phytic acid mutants there is no consistent relationship between phytic acid reduction and mineral concentration. However, I did cite the paper mentioned in its proper context and did discuss this in the text. I note that this reviewer cites one paper that found such a relationship, but didn't mention the half-dozen or so other papers that show no such relationship in low-phytate lines.

5) I included and inhanced discussion of the differences in phytic acid issues between grains and pulses. 

6) I had already included substantial discussion of the question of phytic acid and germination; for example the discussion of the "seed-source effect". I added included discussion of the cited paper. 

7) Re the question of phytic acid and inositol in health issues: I don't think the discussion in the paper is problematic at all. I state for example that there is good science behind the idea that dietary phytic acid may function as an anti-oxident in the gut, and I mention the possible health benefits for inositol. That is not the issue. Titling a paper to suggest that "phytic acid is absorbed and transported" when in fact it isn't is one point I make. That phytic acid is broken down in the gut to yield inositol and that inositol is then taken up and transported is in fact not at all what many of the papers addressing this have said. How can dietary phytic acid contribute to kidney stone formation, if it is in fact not transported to the kidneys? There are serious problems with the contention that phytic acid is absorbed from the gut and has functions requiring transport via blood to other organs and tissues. I think my comments concerning this are fairly straightforward actually.

8 and 9: I have slightly altered what was Figure 1 to make it more comprehensible. I agree that the paper needed more figures. I have added four. 

Reviewer 2 Report

I really enjoyed reading this paper it brought together multiple decades of work on low phytic acid crops into a single location. Of equal importance it indicated some of the innate issues that arise in the interface of agriculture, breeding, pollution, economics and social justice. While here specifically on phytic acid content the overall issues occur in a range of other breeding programs. 

The paper outlines the development and commercialisation (or lack thereof) of phytic acid mutants in plants by a person intimately involved in the entire process across a 40 year period. Its uniqueness lies in the complete view that encompasses not only the scientific work carried out but how that science interacted with the economics (short vs long term thinking of cost for economists vs breeders), unaccounted for environmental costs (P loss), narrowness of scientific focus (single vs multiple outcomes of a project, human vs animal benefits, differential benefits of main and secondary products) and competition between ideologies for funds leading to over emphases on negatives and positives (yield responses in incomplete breeding projects, exaggerated phytic acid benefits) rather than balanced accounting. Overall it shows scientific development of productive outcomes to be a social process rather than a knowledge based process. This demonstration has considerable interest that extends beyond the scope of just phytic acid breeding. It encompasses the issues of all breeders (and probably agronomists also) attempting to develop new materials with significant social benefit but difficult to capture short term economic benefit when favoured alternatives exist with the reverse tendencies.

As a holistic response it has answered a major question overlooked by narrow focussed scientific papers and in this it encompasses a rare level of originality beyond merely looking at value of a few gene mutants.  The personal style of the author is not the norm within a scientific article but in this instance makes it easy to read and to follow the arguments presented.

The comprehensive nature of the writing including all facets allows the author to posit a set of conclusions based on their interpretation of the evidence. However, as is need in a paper with a significant social/scientific interaction it also provides enough information that readers can posit alternative conclusions where they disagree with the interpretations of the author generating an intellectual discussion of value within the breeding and economic and social communities.

I thus think the paper well written, informative and original and well worth publishing.

Author Response

I thank this reviewer for the very nice and supportive comments. I have tried to edit and rewrite the paper to make it better organized and less self-referential, and thus to do a better job of focusing on the important questions.